# Synergistic Effect of Y Doping and Reduction of TiO_2_ on the Improvement of Photocatalytic Performance

**DOI:** 10.3390/nano13152266

**Published:** 2023-08-07

**Authors:** Xijuan Li, Hongjuan Zheng, Yulong Wang, Xia Li, Jinsong Liu, Kang Yan, Jing Wang, Kongjun Zhu

**Affiliations:** 1College of Materials Science and Technology, Nanjing University of Aeronautics and Astronautics, Nanjing 210016, China; lxjnuaa@163.com (X.L.); lixia170904@nuaa.edu.cn (X.L.); jsliu@nuaa.edu.cn (J.L.); 2State Key Laboratory of Mechanics and Control for Aerospace Structures, Nanjing University of Aeronautics and Astronautics, Nanjing 210016, China; yankang@nuaa.edu.cn (K.Y.); wang-jing@nuaa.edu.cn (J.W.); 3Department of Applied Physics, The Hong Kong Polytechnic University, Hong Kong 999077, China; yulong1.wang@polyu.edu.hk

**Keywords:** photocatalysis, TiO_2_, Y doping, H_2_ reduction, hydrothermal method

## Abstract

Pure TiO_2_ and 3% Y-doped TiO_2_ (3% Y-TiO_2_) were prepared by a one-step hydrothermal method. Reduced TiO_2_ (TiO_2_-H_2_) and 3% Y-TiO_2_ (3% Y-TiO_2_-H_2_) were obtained through the thermal conversion treatment of Ar-H_2_ atmosphere at 500 °C for 3 h. By systematically comparing the crystalline phase, structure, morphological features, and photocatalytic properties of 3% Y-TiO_2_-H_2_ with pure TiO_2_, 3% Y-TiO_2_, and TiO_2_-H_2_, the synergistic effect of Y doping and reduction of TiO_2_ was obtained. All samples show the single anatase phase, and no diffraction peak shift is observed. Compared with single-doped TiO_2_ and single-reduced TiO_2_, 3% Y-TiO_2_-H_2_ exhibits the best photocatalytic performance for the degradation of RhB, which can be totally degraded in 20 min. The improvement of photocatalytic performance was attributed to the synergistic effect of Y doping and reduction treatment. Y doping broadened the range of light absorption and reduced the charge recombination rates, and the reduction treatment caused TiO_2_ to be enveloped by disordered shells. The remarkable feature of reduced TiO_2_ by H_2_ is its disordered shell filled with a limited amount of oxygen vacancies (O_V_s) or Ti^3+^, which significantly reduces the E_g_ of TiO_2_ and remarkably increases the absorption of visible light. The synergistic effect of Y doping, Ti^3+^ species, and O_V_s play an important role in the improvement of photocatalytic performances. The discovery of this work provides a new perspective for the improvement of other photocatalysts by combining doping and reduction to modify traditional photocatalytic materials and further improve their performance.

## 1. Introduction

Among the various semiconductors, titanium dioxide (TiO_2_) is considered a useful photocatalyst for the treatment of water pollution and water splitting, owing to its minimal toxicity, strong oxidation capacity, and ready availability [1,2,3,4,5,6]. In recent decades, many research articles have reported the performance, synthesis methods, as well as the reaction mechanisms of TiO_2_ in photocatalytic systems [7,8,9]. The crystalline phase of TiO_2_ remarkably affects its photocatalytic activity. Rutile, anatase, and brookite are the three main crystalline phases of TiO_2_ [10]. The photocatalytic performance of pure anatase TiO_2_ is better than that of rutile, whereas the brookite phase is unstable and does not show appreciable photoactivity [11]. However, the wide bandgap (E_g_; anatase, ~3.20 eV) limits light absorption to the UV region of the solar spectrum (~4% of the total solar irradiance), thus affecting photocatalytic activity [12]. Moreover, the high recombination of the photogenerated electron–hole (e^−^–h^+^) pair has been proven to reduce photocatalytic efficiency [13]. Many approaches, including metal (Al^3+^, Zn^2+^, Fe^3+^, Ce^3+^, Mn^2+^, Ag^+^) [14,15,16] and nonmetal doping [17], the deposition of noble metal (Au, Pt) [18,19], and construction of heterojunctions with other semiconductors [20,21], have been explored to overcome these limitations.

Among these methods, rare earth (RE) metal doping is a popular technique to reduce the recombination rate of photogenerated carriers and shift the absorption wavelength to the visible region and increase the photoactivity of TiO_2_ [22,23]. Furthermore, RE metal doping shows other benefits, such as the ability to concentrate pollutants at the TiO_2_ photocatalyst surface, inhibit the phase transformation from anatase to rutile, and decrease the crystallite size [24,25]. Many studies reported that the concentration of RE metal dopant affects the photoactivity, with the optimum achieved at concentrations below 5 wt.% [25]. Among all the RE ions, yttrium (Y) ion is considered a typical dopant used to modify the electronic structure and optical properties of TiO_2_ [26]. Specifically, it has been found that the Y-doped TiO_2_ can reduce the recombination rate of photogenerated electrons/holes pairs, which improves the photocatalytic efficiency of TiO_2_ [27]. The effect of Y-doped TiO_2_ on photocatalytic activity has been recently reported in several studies [28,29].

In addition to RE metal doping, reduced TiO_2_, which is obtained by high-temperature treatments in various reducing atmospheres (e.g., vacuum, Ar, Ar-H_2_, and pure H_2_) [30,31,32] or thermite reduction [33], reduction with sodium borohydride [34], and Ti^3+^ self-doping [35,36], has shown tremendous potential as a photocatalyst in wastewater treatment and water splitting recently. The enhanced photocatalytic performance of reduced TiO_2_ is related to a significant decrease in E_g_ and increased light absorption due to the introduction of oxygen vacancies (O_V_) or the formation of Ti^3+^ centers in the TiO_2_ lattice [34]. Current studies on reduced TiO_2_ focused on the relationship between the Ti^3+^ or O_V_ concentration and photocatalytic activity of TiO_2_ and comparison of the reduction degree caused by different reduction methods. However, the synergistic effect of RE doping and reduction treatment for TiO_2_ photocatalysts has not been reported yet.

In this work, pure anatase TiO_2_, Y-doped TiO_2_, and their respective reduced products were prepared to verify the synergistic effect of Y doping and reduction treatment for TiO_2_ photocatalysts. Pure anatase TiO_2_ and Y-doped TiO_2_ were obtained by a one-step hydrothermal method, reduced TiO_2_ and reduced Y-doped TiO_2_ were formed by heat treatment in an Ar-H_2_ atmosphere at 500 °C for 3 h. The crystalline phase, morphological features, and photocatalytic properties of four types of TiO_2_ were compared systematically. By comparing TiO_2_ and Y-doped TiO_2_, the effect of RE doping on the improvement of photocatalytic performance could be obtained. Similarly, by comparing TiO_2_ with reduced TiO_2_, the effect of reduction treatment by H_2_ on the improvement of photocatalytic performance can be obtained. By comparing TiO_2_, which are simultaneously doped and reduced, with pure TiO_2_, Y-doped TiO_2_, and reduced TiO_2_, it can be verified the existence of a coupling effect of RE doping and reduction treatment on the improvement of photocatalytic performance. Ultimately, a positive effect of the combination of Y doping and reduction of TiO_2_ on the improvement of photodegradation activity of organic pollutants under simulated sunlight irradiation was found, and their possible photocatalytic mechanism was proposed. The aim of this study was to develop an understanding of the synergistic effect of Y doping and reduction of TiO_2_ on the improvement of photocatalytic performance.

## 2. Experimental

### 2.1. Materials

Titanium isopropoxide (TIP), Y(NO_3_)_3_·6H_2_O, absolute ethanol, acetic acid, and nitric acid (HNO_3_) were received from Sinopharm Chemical Reagent Co., Ltd., Shanghai, China. All chemicals were not further purified during use.

### 2.2. Preparation of Pure TiO_2_ and Y-Doped TiO_2_ Nanoparticles

A total of 4.2 mL of TIP, 16.8 mL of absolute ethanol, and 1.68 mL of acetic acid were mixed under continuous stirring to form solution A. Solution B was composed of a certain amount of Y(NO_3_)_3_·6H_2_O dissolved in 21 mL of deionized water with a pH value adjusted to 2.5 by dilute HNO_3_ (1 mol·L^−1^). The molar ratio of Y/Ti in solution B was set to 0, 1%, 2%, 3%, 4%, and 5%, respectively. Then, solution B was added into solution A to obtain Y-doped TiO_2_ sol. After stirring for 30 min, the Y-doped TiO_2_ sol was transferred into a 70 mL Teflon vessel. The Teflon vessel was placed into stainless-steel autoclaves. Then, the autoclaves were put in an oven and heated to 180 °C for 12 h. Then, the autoclaves were cooled to room temperature after hydrothermal treatment. The final products were filtered, washed, and dried at 80 °C for 12 h.

### 2.3. Preparation of Reduced TiO_2_ and Y-Doped TiO_2_ Nanoparticles

TiO_2_ and Y-doped TiO_2_ nanoparticles were placed in a tube furnace, respectively, and then evacuated to a base pressure of about 0.5 Pa. The tube was filled with Ar-H_2_ (95 vol %–5 vol %) atmosphere to normal pressure. The samples were heated at 500 °C with a heating rate of 5 °C /min for 3 h in Ar-H_2_ flow to obtain the final reduced pure TiO_2_ and Y-doped TiO_2_ products by H_2_.

### 2.4. Photocatalytic Test

The photocatalytic activity test was conducted using RhB as a simulated pollutant and a 300 W Xe lamp as a simulated sunlight source (PLS-SXE300/300UV, Trusttech Co., Ltd., Beijing, China) to irradiate the pollutant. In a typical photocatalytic process, 10 mg of the catalyst was dispersed into 100 mL of the RhB solution (10 mg L^−1^) in a 200 mL double-layer reactor cooled by running water to maintain a temperature of 25 °C, and the mixture was magnetically stirred with 300 rpm in the dark for 1 h to achieve an adsorption–desorption balance. Then, the xenon lamp was turned on, and 5 mL of the suspension was collected at every 10 min interval and centrifuged (9000 rpm, 5 min) to remove the catalyst powders. The total irradiation time of all samples was 1 h. The degradation rate of RhB was calculated by converting the absorbance into concentration using the dye standard curve by a UV–vis spectrophotometer.

### 2.5. Characterization

The phase structure of the products was determined using X-ray diffraction (XRD) (Bruker D8Advance diffractometer) with Cu K_α_ radiation (λ = 1.5418 Å). The structural information of products was obtained using high-resolution transmission electron microscopy (HRTEM) by a JEM-2100F at 200 kV. X-ray photoelectron spectroscopy (XPS) measurements were carried out using the AXIS-Ultra DLD system. The UV–vis diffuse reflectance absorption spectra (DRS) of samples were used to determine the bandgap by the PE Lambda 950 spectrometer with BaSO_4_ as reference. The Brunauer–Emmett–Teller (BET) method with a JW-BK200B was used to evaluate the specific surface area of the products.

## 3. Results and Discussion 

### 3.1. Structure and Morphology Characterization

Pure TiO_2_ and various Y-doped TiO_2_ (1–5%) were prepared in advance to verify their phase composition and photocatalytic performance for the degradation of RhB. XRD was used to characterize and compare the crystalline phase of samples. All samples show the single anatase phase (Appendix A). The catalytic activities were evaluated by the degradation of RhB under simulated sunlight irradiation and the maximum photocatalytic performance for the degradation of RhB when the Y dopant concentration was 3% (Appendix A). Therefore, 3% Y-doped TiO_2_ was selected as a typical research object for rare earth doping. Figure 1 shows the crystalline phase of pure TiO_2_, 3% Y-doped TiO_2_ (3% Y-TiO_2_), reduced TiO_2_ by H_2_ (TiO_2_-H_2_), and reduced 3% Y-doped TiO_2_ (3% Y-TiO_2_-H_2_). All four samples have significant diffraction peaks representing the characteristic of a single anatase phase (PDF No. 211272). Anatase is the only crystalline phase present in the structure of TiO_2_-H_2_ and 3% Y-TiO_2_-H_2_, indicating that no phase transformation occurs in the annealing process of H_2_ reduction at 500 °C. After annealing, the colors of TiO_2_-H_2_ and 3% Y-TiO_2_-H_2_ change from white to gray black. The refraction of Y_2_O_3_ is not observed in the XRD patterns of 3% Y-TiO_2_ or 3% Y-TiO_2_-H_2_, indicating that the content of Y_2_O_3_ is below the detection limit. No diffraction peak shift is observed for all Y-modified samples, demonstrating that Y^3+^ species exists at the crystal boundary or surface rather than in the inner crystalline structure of TiO_2_. The increased diffraction peak intensities of TiO_2_-H_2_ and 3% Y-TiO_2_-H_2_ after annealing indicate the increased crystallinity of samples in comparison with those of pure TiO_2_ and 3% Y-TiO_2_.

The morphology and structure of all four samples are investigated by HRTEM. The low-magnification TEM images and statistical particle size distribution of TiO_2_, 3% Y-TiO_2_, TiO_2_-H_2_, and 3% Y-TiO_2_-H_2_ are shown in Figure 2a–d and their insert, respectively. All samples are primarily composed of dispersed circular, rectangular, and some irregular-shaped nanoparticles with average sizes ranging from 8 nm to 11 nm. The grain sizes of TiO_2_-H_2_ and 3% Y-TiO_2_-H_2_ increase slightly but not significantly after annealing. High-magnification TEM images show clear lattice fringes, which is indicative of the high crystallinity of TiO_2_ and 3% Y-TiO_2_ (Figure 2e,f). However, high-magnification TEM images of TiO_2_-H_2_ and 3% Y-TiO_2_-H_2_ illustrate a core–shell structure with a ∼1.5 nm-thick disordered surface shell (Figure 2g,h), which is not observed on the surfaces of TiO_2_ and 3% Y-TiO_2_ prepared by the single hydrothermal method. Four samples have lattice spacings of 0.34 nm in the same way as in Figure 2e, which is consistent with its (101) planes (Figure 2e–h). The elemental mapping images of 3% Y-TiO_2_-H_2_ with individual elements of Ti, O, and Y are shown in Figure 2i. Ti, O, and Y are uniformly distributed throughout the particle space, proving the presence of Y elements in Y-doped TiO_2_ samples.

The surface areas of all products were estimated by the BET analysis; their N_2_ adsorption–desorption isotherms are shown in Figure 3. The specific surface areas of pure TiO_2_ and 3% Y-TiO_2_ are 148.58 (Figure 3a) and 160.17 m^2^/g (Figure 3b), respectively, which can be related to changes in the morphology of the TiO_2_ after doping with Y. Based on the XRD results, no diffraction peak shift is observed for 3% Y-doped TiO_2_, demonstrating that the Y^3+^ species exists at the crystal boundary or surface rather than in the inner crystalline structure of TiO_2_. Therefore, it is speculated that the adhesion of Y^3+^ species to the surface of particles increases the roughness of the particle surface, resulting in an increase in the specific surface area. The specific surface areas of TiO_2_-H_2_ and 3% Y-TiO_2_-H_2_ are 85.94 (Figure 3c) and 117.42 m^2^/g (Figure 3d), respectively, which is lower compared to pure TiO_2_ and 3% Y-TiO_2_. This may be attributed to the grain growth and slight aggregation caused by high-temperature annealing treatment, which is consistent with the results of the statistical particle size distribution in TEM images. 

Based on the survey scanning the XPS spectra of four samples, all labeled peaks were attributed to Ti 2p and O 1s (Appendix A). Furthermore, a weak Y 3d peak is observed at ~158 eV, indicating the presence of Y^3+^ species on the surface of 3% Y-TiO_2_ and 3% Y-TiO_2_-H_2_ samples. The high-resolution XPS spectra of Ti 2p for four samples are shown in Figure 4a–d. TiO_2_ and 3% Y-TiO_2_ exhibit two peaks at 458.4 and 464.2 eV, which are attributed to 2p3/2 and 2p1/2 of Ti^4+^, respectively [37]. For TiO_2_-H_2_ and 3% Y-TiO_2_-H_2_, these peaks shift to low values, broaden, and become unsymmetrical compared with those of pure TiO_2_ and 3% Y-TiO_2_, indicating a different bonding environment (Appendix A) [38]. The fitting curves show that the two peaks of Ti 2p are divided into four peaks, found at 458.3 and 464.0 eV, respectively, corresponding to the 2p3/2 and 2p1/2 peaks of Ti^4+^, and at 457.8 and 463.2 eV, respectively, corresponding to the 2p3/2 and 2p1/2 peaks of Ti^3+^ [39]. The Ti^3+^ state indicates that TiO_2_-H_2_ and 3% Y-TiO_2_-H_2_ are partially reduced through H_2_ reduction. 

The high-resolution XPS spectra of O 1s for four samples are shown in Figure 4e–h. In TiO_2_ and 3% Y-TiO_2_, two nonsymmetric fitted peaks signify the presence of two O species. The binding energy at 529.6 eV is ascribed to the characteristic peak of Ti-O in anatase TiO_2_, namely, lattice oxygen (O_L_). The binding energy at 531.4 eV is attributed to the O-H, i.e., adsorbed oxygen (O_A_) [40]. In TiO_2_-H_2_ and 3% Y-TiO_2_-H_2_, three nonsymmetric fitted peaks are obtained. Except O_L_ (529.8 eV) and O_A_ (531.9 eV), the small peak centered at around 530.9 eV can be assigned to O_V_s [41]. The estimated ratio of O_V_ to O_L_ (O_V_/O_L_) of TiO_2_-H_2_ is 0.09 according to the calculated integral areas of the corresponding peaks. For 3% Y-TiO_2_-H_2_, the ratio of O_V_/O_L_ decreases to 0.07, indicating that the Y^3+^ on the surface or grain boundary of TiO_2_ by Y doping may slightly inhibit the formation of O_V_s. Furthermore, the content of O_A_ decreases sharply for TiO_2_-H_2_ and 3% Y-TiO_2_-H_2_, implying that -OH groups or adsorbed water on the surface of TiO_2_-H_2_ and 3% Y-TiO_2_-H_2_ are largely scavenged during annealing.

Figure 5a shows the UV–vis DRS of four samples. The light absorption edge of pure TiO_2_ is approximately 390 nm. The light absorption edges of 3% Y-TiO_2_, TiO_2_-H_2_, and 3% Y-TiO_2_-H_2_ gradually blue-shift compared with that of TiO_2_, and the light absorption capacities in their visible region are gradually enhanced. The light absorption in the region from ~400 nm to the near-infrared region is significant in TiO_2_-H_2_ and higher in 3% Y-TiO_2_-H_2_.

The E_g_ can be calculated in accordance with the formula: (1)$${\left(\alpha hv\right)}^{1/n}=A\left(hv-E_{g}\right),$$

Among them, α, *h*, ν, and *A* are the absorption coefficient, Planck’s constant, frequency of the incident light, and a constant, respectively. For direct and indirect transition semiconductors, *n* is 1/2 and 2, respectively. The value of *n* is 2 for the anatate TiO_2_. In Figure 5b, the E_g_ values of TiO_2_, 3% Y-TiO_2_, TiO_2_-H_2_, and 3% Y-TiO_2_-H_2_ obtained from the tangent intercept are 2.88, 2.70, 2.24, and 2.17 eV, respectively. 

### 3.2. Photocatalytic Test

The effect on the catalytic activities of all four samples is evaluated by the degradation of RhB under simulated sunlight irradiation, as shown in Figure 6. The UV–vis absorption spectra of TiO_2_, 3% Y-TiO_2_, TiO_2_-H_2_, 3% Y-TiO_2_-H_2_ are separately displayed in Figure 6a–d, and the degradation efficiency curves are summarized in Figure 6e. For pure TiO_2_, RhB can be totally degraded in approximately 60 min, whereas RhB can be totally degraded by 3% Y-TiO_2_ in 40 min, which is shorter than that by pure TiO_2_. This result indicates that a proper amount of RE doping has a positive effect on photocatalytic performance. For TiO_2_-H_2_, RhB can be totally degraded only in 30 min, which is beneficial for the reduction role of TiO_2_. Interestingly, RhB can be totally degraded by 3% Y-TiO_2_-H_2_ in 20 min, which is less time than those of other samples. The combined action of 3% Y doping and reduction by H_2_ is beneficial to the photodegradation activity of TiO_2_, and the co-modified catalyst exhibits higher photocatalytic activity than any single-modified catalyst. 

The kinetics of the photocatalytic activities of all four samples follow the first-order reaction:−ln(*C*_t_/*C*_0_) = *k*_1_t, (2)
where *k*_1_ is the pseudo-first-order reaction rate constant (min^−1^) obtained from the slope of −ln(*C*_t_/*C*_0_) vs. t, as shown in Figure 6f,g. The *k*_1_ values of TiO_2_, 3% Y-TiO_2_, TiO_2_-H_2_, and 3% Y-TiO_2_-H_2_ are 0.0606, 0.0985, 0.1269, and 0.1746, respectively, indicating the efficient photodegradation activity of 3% Y-TiO_2_-H_2_. Therefore, the combination of Y doping and H_2_ reduction is responsible for the high degradation rate.

### 3.3. Mechanism Analysis

RE metal doping can significantly modify the electrical, physical, and chemical properties of TiO_2_ photocatalyst and play an effective role for improving photocatalytic performance. Compared to pure TiO_2_, 3% Y-doped TiO_2_ has a faster degradation rate of RhB, which is attributed to the influence of Y doping. The ionic radius of Y^3+^ (93 pm) is larger than that of Ti^4+^ (68 pm), which is difficult to replace Ti into the TiO_2_ lattice directly [25]. Based on the XRD results, no diffraction peak shift is observed for 3% Y-doped TiO_2_, demonstrating that Y^3+^ species exists at the crystal boundary or surface rather than in the inner crystalline structure of TiO_2_. Furthermore, the survey scanning the XPS spectra (Appendix A) shows that a weak Y 3d peak is observed at ~158 eV, indicating the presence of Y^3+^ species on the surface of 3% Y-TiO_2_. Based on XRD and XPS, the Y^3+^ species might be deposited on the surface of TiO_2_. Given that the work function (Φ) of RE metals is lower than that of titanium, RE^3+^ has more tendencies to attract the e^−^ from the sample surface compared with Ti ions, resulting in a reduced e^−^–h^+^ pair recombination rate. In addition, 3% Y-TiO_2_ with a higher specific surface area has more reactive active sites compared with pure TiO_2_, which also helps to improve the photocatalytic performance. Moreover, UV–vis DRS revealed that 3% Y-TiO_2_ has higher light absorption capacities in their visible region than pure TiO_2_, resulting in lower E_g_ values. However, an excessive amount of metal dopants may lead to a decline in photodegradation efficiency towards pollutants by reducing the yield of photoinduced e^−^–h^+^ pairs (Appendix A). Therefore, 3% Y doping is the most suitable doping amount and selected as the subsequent research object in our work. 

The reduction treatment of TiO_2_ by H_2_ is also one of the important methods to improve its photocatalytic performance. Compared to pure TiO_2_ and 3% Y-TiO_2_, the specific surface areas of TiO_2_-H_2_ significantly decreased, indicating that the reaction active sites decreased. However, TiO_2_-H_2_ also had a faster degradation rate of RhB, which is attributed to the changes in microstructure caused by reduction treatment. The remarkable feature of reduced TiO_2_ by H_2_ is its disordered shell filled with a limited amount of O_V_s or Ti^3+^, which is consistent with the HRTEM and XPS results. Furthemore, the significant decrease in E_g_ values of TiO_2_-H_2_ is due to the significant increase in the sample’s absorption capacity for visible light according to UV–vis DRS results. Compared with the RE metal doping, O_V_ or Ti^3+^ is a kind of self-doping of the crystal itself without introducing any impurity element, which is considered to reflect the influence of the internal modification of TiO_2_ on its physicochemical and photocatalytic performance, such as tuning optical absorption, reducing E_g_, and increasing carrier concentration [31].

Under the simultaneous action of Y doping and reduction treatment, 3% Y-TiO_2_-H_2_ exhibited a higher photocatalytic degradation ability than any single-modified catalyst. Thus, a possible photocatalytic mechanism for 3% Y-TiO_2_-H_2_ nanoparticles is proposed (Figure 7). The disordered shell of reduced 3% Y-TiO_2_-H_2_ is filled with a limited amount of O_V_s and Ti^3+^, which significantly reduces the E_g_ of TiO_2_ and remarkably increases the absorption of visible light. The formation of a defective energy level caused by O_V_ and Ti^3+^ below the conduction band of TiO_2_ decreases the excitation energy, resulting in highly active photocatalysts [42]. The incorporation of Y^3+^ on the surface of TiO_2_ increases the charge separation by acting as an electron trapper, consequently produces more e^−^ for reaction on the surface of the catalyst, and reduces the e^−^-h^+^ recombination rate of photocatalyst. In addition to reducing carrier recombination, as a member of RE elements, Y doping broadens the range of light absorption and increases the photoactivity of TiO_2_; this phenomenon is consistent with the DRS and photodegradation results. In conclusion, the more reactive active sites, the rapid interfacial charge transfer, and stronger optical absorption of 3% Y-TiO_2_-H_2_ caused by Y doping should be the reason for the higher rate of photocatalytic reactions than those of undoped TiO_2_-H_2_ [43]. Their photocatalytic process is such that, under simulated sunlight irradiation, e^−^ is excited from VB to CB of 3% Y-TiO_2_-H_2_, and h^+^ in VB is abandoned. Generated charge carriers react with oxygen molecules, water molecules, or OH^−^ to produce oxidative species for the degradation of organic dye in the aqueous solution, such as hydroxyl radicals (•OH) and (•O_2_^−^).

## 4. Conclusions

Y doping and the reduction treatment of TiO_2_ have been widely proven to be an effective way to improve their photocatalytic performance, respectively. In this work, the coupling treatment of Y doping and reduction of TiO_2_ can further improve the photocatalytic performance, which is better than any individual method. In summary, pure TiO_2_ and 3% Y-TiO_2_ were prepared by a one-step hydrothermal method. Reduced TiO_2_-H_2_ and 3% Y-TiO_2_-H_2_ were obtained through the thermal conversion treatment of Ar-H_2_ atmosphere at 500 °C for 3 h. All samples show the single anatase phase, and no diffraction peak shift is observed. The photodegradation efficiency of pure TiO_2_, 3% Y-TiO_2_, TiO_2_-H_2_, and 3% Y-TiO_2_-H_2_ on RhB gradually increased. The 3% Y-TiO_2_-H_2_ exhibits the best photocatalytic performance for the degradation of RhB among these four samples, which can be totally degraded in 20 min. The key factors for the improved photocatalytic performance of 3% Y-TiO_2_-H_2_ could be attributed to the synergistic effect of Y doping and the reduction of TiO_2_. Y metal-ion doping broadened the range of light absorption and reduced the charge recombination rates. Reduction treatment can cause TiO_2_ to be enveloped by disordered shells, and O_V_s and Ti^3+^ species efficiently reduced the E_g_ of TiO_2_, remarkably increasing the absorption of light. The synergistic effect of rapid interfacial charge transfer and the stronger optical absorption of 3% Y-TiO_2_-H_2_ caused by Y doping and reduction treatment should be the reason for the high photocatalytic efficiency. The discovery of this work provides a new perspective for the improvement of other photocatalysts by combining doping and reduction to modify traditional photocatalytic materials and further improve their performance.

## Figures and Tables

**Figure 1 nanomaterials-13-02266-f001:**
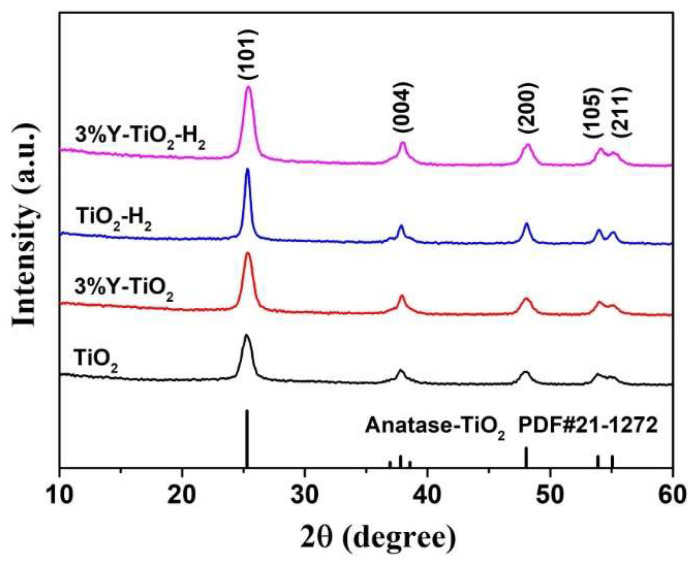
XRD patterns of pure TiO_2_, 3% Y-TiO_2_, TiO_2_-H_2_, and 3% Y-TiO_2_-H_2_ samples.

**Figure 2 nanomaterials-13-02266-f002:**
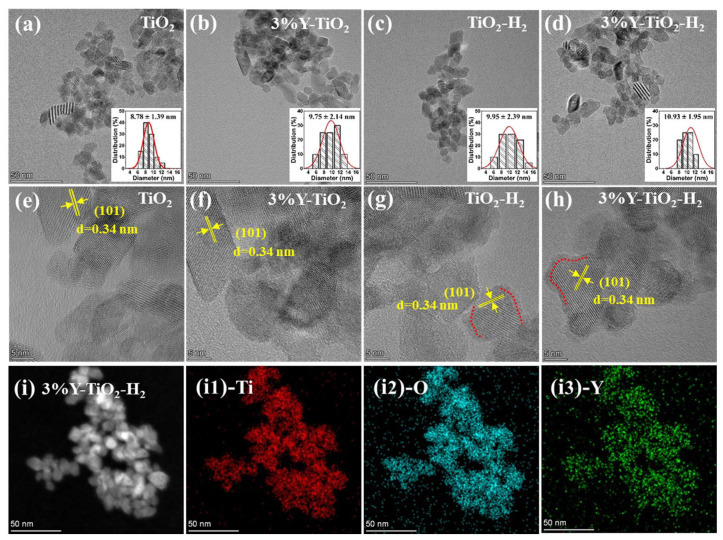
HRTEM images of (**a**,**e**) TiO_2_, (**b**,**f**) 3% Y-TiO_2_, (**c**,**g**) TiO_2_-H_2_, and (**d**,**h**) 3% Y-TiO_2_-H_2_ (inset is the corresponding statistical particle size distribution). (**i**) Elemental mapping of 3% Y-TiO_2_-H_2_ for Ti (**i1**), O (**i2**), and Y (**i3**).

**Figure 3 nanomaterials-13-02266-f003:**
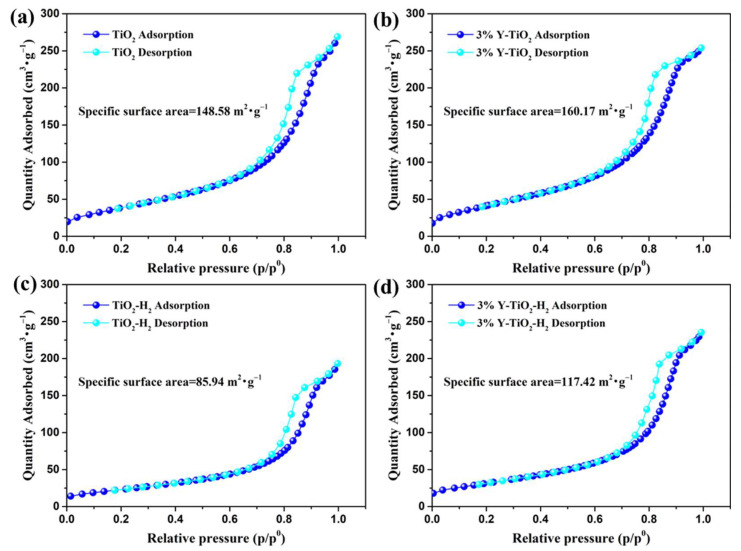
N_2_ adsorption–desorption isotherms of TiO_2_ (**a**), 3% Y-TiO_2_ (**b**), TiO_2_-H_2_ (**c**), and 3% Y-TiO_2_-H_2_ (**d**).

**Figure 4 nanomaterials-13-02266-f004:**
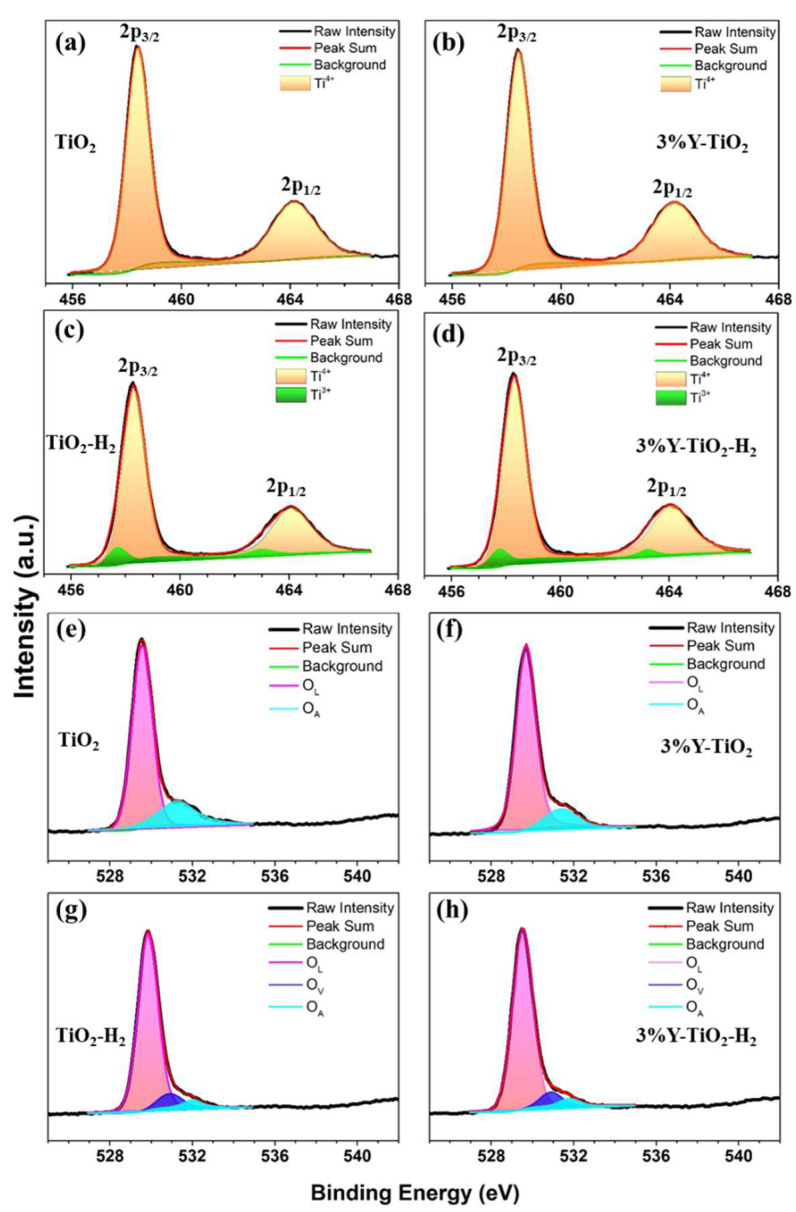
XPS spectra for Ti 2p of TiO_2_ (**a**), 3% Y-TiO_2_ (**b**), TiO_2_-H_2_ (**c**), 3% Y-TiO_2_-H_2_ (**d**), and for O 1s of TiO_2_ (**e**), 3% Y-TiO_2_ (**f**), TiO_2_-H_2_ (**g**), 3% Y-TiO_2_-H_2_ (**h**).

**Figure 5 nanomaterials-13-02266-f005:**
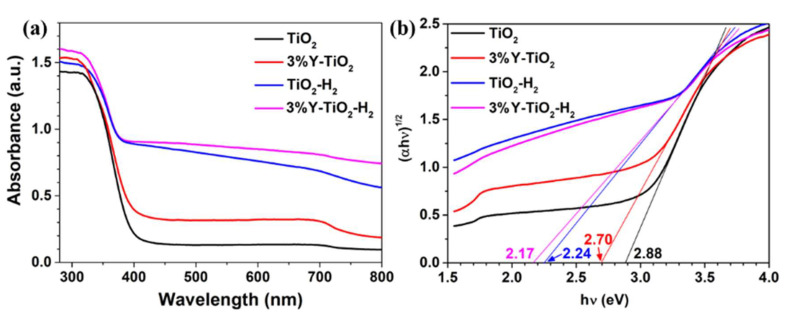
(**a**) UV–vis diffuse reflectance absorption spectra and (**b**) corresponding bandgap of TiO_2_, 3% Y-TiO_2_, TiO_2_-H_2_, and 3% Y-TiO_2_-H_2_ samples.

**Figure 6 nanomaterials-13-02266-f006:**
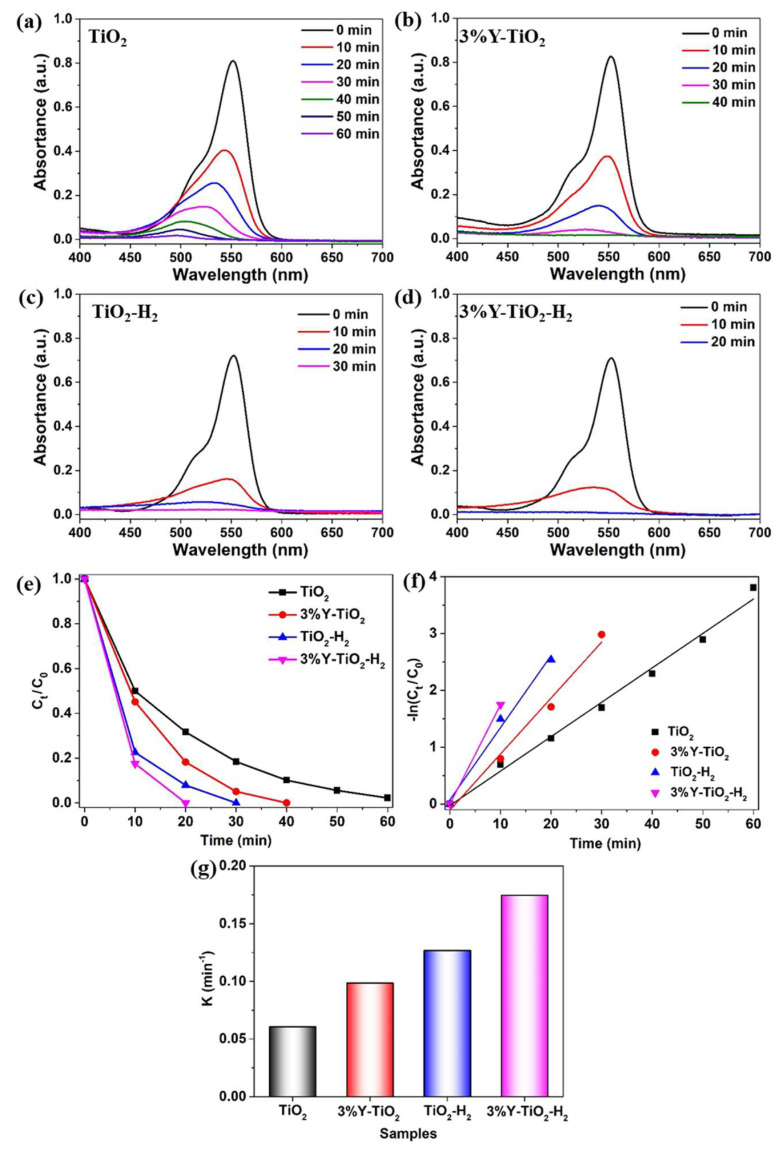
UV–vis absorption spectra of TiO_2_ (**a**), 3% Y-TiO_2_ (**b**), TiO_2_-H_2_ (**c**), 3% Y-TiO_2_-H_2_ (**d**), degradation efficiency curves (**e**), kinetic curves (**f**), and reaction rate constant (**g**).

**Figure 7 nanomaterials-13-02266-f007:**
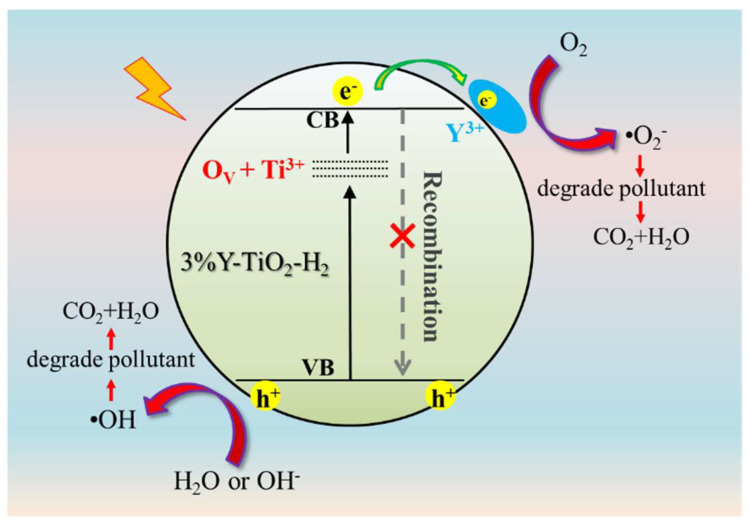
Schematic of the photodegradation of RhB in 3% Y-TiO_2_-H_2_ nanoparticles.

## Data Availability

All data generated or analyzed during this work are included in this published article.

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
