# Peer review of "Synergistic Effect of Y Doping and Reduction of TiO2 on the Improvement of Photocatalytic Performance"

_nanomaterials, 2023, doi:10.3390/nano13152266_

Round 1
Reviewer 1 Report
In this work, 3% Y-doped TiO2 (3% Y-TiO2), reduced TiO2 by H2 (TiO2-H2), and reduced 15 3% Y-TiO2 by H2 (3% Y-TiO2-H2), were prepared and characterized and its photocatalytic performance was investigated in RhB photoreduction. Readers in this field may find the paper interesting, but there are still some issues to be addressed. Please see the following comments:
- The abstract should be more quantitative.
- Lines 17-18: “… degrades 10 mg 17 L−1 rhodamine B (RhB) …”: It doesn’t mean the efficiency of degradation and experimental conditions (i.e. pH, temperature, time, and photocatalyst dosage) should be reported.
- The novelty of the work is unclear and should be highlighted
- The literature review should be improved
An overview of the metal ions used to dope TiO2 for improved photocatalytic activity (e.g., Chemosphere 281, 130795) should be included in the literature review.
- Why was Y chosen for doping TiO2?
- Photocatalytic performance should be reported after photocatalyst characterization
- What is the evidence for the proposed mechanism?
- What is the role of surface area in TiO2 photocatalytic behavior after its doping/reduction?
- What is the role of Y% on the Eg value of TiO2?
- How was the Y leaching within the photocatalytic process?
- All the reaction conditions must be added to the caption of figure 3.
Reviewer 2 Report
In my opinion, in order to improve the manuscript, it is desirable to address the following comments:
1. Clearly define the purpose of your work both in Abstract and at the end of Introduction;
2. In Conclusion, indicate the novelty of your work, taking into the results obtained by other authors;
3. Abstract is a short summary of the article. From Abstract, the reader should understand the essence of the study. Your Abstract, beginning with: "Y doping can reduce the charge..." (lines 20-28), outlines your assumptions. It is desirable that this part of the text describes the obtained results.
4. After reading the first sentence of Introduction: "Among semiconductors, titanium dioxide (TiO2) is considered a useful photocatalyst for the treatment of water pollution or water splitting owing to its minimal toxicity, strong oxidation capacity, and ready availability [1, 2]" (lines 35 -37), the reader might think that, since 2007 [1], no one has been working on titanium oxide. This impression is strengthened by the absence of articles on the chosen topic in References for the period of 2021-2023. While a search for the keywords "TiO2 and photocatalysis", for example, in https://www.sciencedirect.com/, showed that only in 2023 these terms were used in more than 1000 articles. Please update the review part of the article.
5. If Figure 4a shows UV-vis diffuse reflectance spectra, then the y-axis should be named Reflectance.
6. What experiments suggested that: "Y metal ions existed on the TiO2 surface, reduced the charge recombination rates..." (lines 236-238)?
Reviewer 3 Report
I have had the opportunity to review your manuscript titled "Synergistic effect of Y doping and reduction of TiO2 on the improvement of photocatalysis" and I appreciate your efforts in investigating the improved photocatalytic performance of TiO2 through Y doping and reduction. Your research is of immense interest, and it will undoubtedly contribute to the development of novel TiO2-based photocatalysts.
Upon careful consideration, I have a few points for further clarification and minor revision:
- Title: The title of the paper does well in summarizing the main points of your study. However, considering the emphasis of your research on enhanced photocatalytic performance, it would be beneficial to include this aspect in the title.
- Abstract: The abstract is well-written and provides a concise summary of the work. However, I recommend that you provide more quantitative results in the abstract to better convey your findings to readers.
- Methodology: Could you provide more details about the preparation process of your samples? This would help in understanding the exact methodology and allow other researchers to replicate your work.
- Results and Discussion: You state that 3% Y-TiO2-H2 has superior photocatalytic activity compared to single-doped or single-reduced TiO2. However, it would be beneficial to see a more direct comparison or quantitative analysis to support this claim.
- Mechanism: You have mentioned that Y doping reduces the charge recombination rates and acts as a charge trapper for photogenerated carriers. Could you please elaborate on this mechanism, possibly with the help of a diagram or schematic representation?
- Synergistic Effect: While you mention the synergistic effect of Y doping, Ti3+ species, and oxygen vacancies, it would be helpful to understand more about their individual contributions and how they interact to improve photocatalytic performance.
- Conclusion: The conclusions drawn from your study could be strengthened by tying them more explicitly to the findings and data presented earlier in the paper.
I hope you find these comments helpful. Your work is certainly promising and I believe that with these revisions, it can contribute significantly to the existing body of literature on this topic.
Round 2
Reviewer 1 Report
The authors well-revised the paper, but the response to Q#8 of Reviewer #1 about the effect of surface area on photocatalytic performance is not convincing. Surface area can be an important parameter and cannot be distinguished by TEM images.
Author Response
"Please see the attachment.

Reviewer 2 Report
Comments on the authors' answers. Below are just my questions.
1. Clearly define the purpose of your work both in Abstract and at the end of Introduction.
Disagree. Clearly define the purpose of your work in Abstract
2. In Conclusion, indicate the novelty of your work, taking into the results obtained by other authors;
Disagree. In text missing the results obtained by other authors.
3. Abstract is a short summary of the article. From Abstract, the reader should understand the essence of the study. Your Abstract, beginning with: "Y doping can reduce the charge..." (lines 20- 28), outlines your assumptions. It is desirable that this part of the text describes the obtained results.
Disagree. Clearly define the purpose of your work in Abstract.
4. After reading the first sentence of Introduction: ...
Disagree. Please indicate new articles in the References
5. If Figure 4a shows UV-vis diffuse reflectance spectra, then the y-axis should be named Reflectance.
Agree.
6. What experiments suggested that: "Y metal ions existed on the TiO2 surface, reduced the charge recombination rates..." (lines 236-238)?
Agree.
Author Response
"Please see the attachment.
